# Gaussian Kernel Mixture Network for Single Image Defocus Deblurring

**Yuhui Quan**[*],     **Zicong Wu**[*]
School of Computer Science and Engineering
South China University of Technology

**Hui Ji**[*]
Department of Mathematics
National University of Singapore

## Abstract

Defocus blur is one kind of blur effects often seen in images, which is challenging to remove due to its spatially variant amount. This paper presents an end-to-end deep learning approach for removing defocus blur from a single image, so as to have an all-in-focus image for consequent vision tasks. First, a pixel-wise Gaussian kernel mixture (GKM) model is proposed for representing spatially variant defocus blur kernels in an efficient linear parametric form, with higher accuracy than existing models. Then, a deep neural network called GKMNet is developed by unrolling a fixed-point iteration of the GKM-based deblurring. The GKMNet is built on a lightweight scale-recurrent architecture, with a scale-recurrent attention module for estimating the mixing coefficients in GKM for defocus deblurring. Extensive experiments show that the GKMNet not only noticeably outperforms existing defocus deblurring methods, but also has its advantages in terms of model complexity and computational efficiency.

## 1   Introduction

The appearance sharpness of an object in an image taken by a camera is determined by the scene distance of the object to the focal plane of the camera. An object will have the sharpest appearance when it is on the focal plane, *i.e.*, the object is in focus. The area around the focal plane where objects appear to be in focus is called the depth of field (DoF). When an object is away from the DoF, it will appear blurry. The further is an object away from the DoF, the more blurry it appears. Such a phenomenon is called defocus blur or out-of-focus blur. Defocus blur effects will be prominent in an image with a shallow DoF, *e.g.* images captured with a large aperture. This paper concerns the problem of single image defocus deblurring (SIDD) which is about reconstructing an all-in-focus image from a defocused image (*i.e.* an image with defocused regions). SIDD is of practical values to many applications in machine vision, *e.g.*, photo refocusing, object recognition, and many others [1].

Consider a defocused image $\boldsymbol{y}$, which relates to its all-in-focus counterpart $\boldsymbol{x}$ by

$$\boldsymbol{y} = \boldsymbol{B} \circ \boldsymbol{x} + \boldsymbol{\epsilon}, \tag{1}$$

where $\boldsymbol{\epsilon}$ denotes the measurement noise, and $\boldsymbol{B}$ is a linear operator defined by

$$(\boldsymbol{B} \circ \boldsymbol{x})[m,n] := \sum_i \sum_j \boldsymbol{b}_{m,n}[i,j]\boldsymbol{x}[m-i, n-j]. \tag{2}$$

Each pixel at location $[m,n]$ is associated with a defocus kernel $\boldsymbol{b}_{m,n}$, also referred to as point spread function (PSF), which is determined by the distance to the focal plane. Often, these pixel-wise PSFs are approximated by Gaussian kernels [2, 3, 4, 5, 6] or disk kernels [7, 8]. Without supplementary information on the scene depth, these pixel-wise PSFs are unknown. Therefore, SIDD is a challenging nonlinear inverse problem which needs to estimate both $\boldsymbol{B}$ and $\boldsymbol{x}$ from (1).

---

[*]email: csyhquan@scut.edu.cn (Y. Quan); cszicongwu@mail.scut.edu.cn (Z. Wu); matjh@nus.edu.sg (H. Ji).

35th Conference on Neural Information Processing Systems (NeurIPS 2021).

## 1.1 Discussion on Existing Work

Most existing methods (*e.g.* [2, 7, 4, 8, 5, 6, 9]) take a two-stage approach which *(i)* estimates a dense defocus map to derive the operator $\boldsymbol{B}$ and then *(ii)* recovers the image $\boldsymbol{x}$ by using nonblind image deconvolution to solve (1) with the estimated $\boldsymbol{B}$. Generally, such a two-stage approach has a long pipeline with many modules, and the estimation error in one module will be magnified in the consequent modules. For instance, the defocus amount only can be estimated on a subset of image pixels such as edge points. Then a dense defocus map needs to be constructed by propagating these few estimations to all pixels. It can be seen that any error in the sparse defocus map will result in erroneous PSFs, and unfortunately deconvolution is very sensitive to the errors in PSFs [10, 11]. As a result, the two-stage approach does not perform well in practice. Also, the computational cost in the second stage is high for a non-uniform blurring operator $\boldsymbol{B}$, as the inversion process regrading $\boldsymbol{B}$, which is often called many times in nonblind image deconvolution, cannot be efficiently computed via fast Fourier transform (FFT).

Deep learning has become one prominent tool for solving a wide range of image restoration problems. In comparison to the rapid progress of DNNs for spatially-varying motion deblurring (*e.g.* [12, 13, 14, 15, 16, 17]), there have been few works on studying DNN-based approaches to defocus deblurring. One might directly adapt an existing motion deblurring DNN for SIDD. However, the kernels of defocus blur are very different from those of motion blur, *e.g.*, roughly isotropic support vs highly curvy support. Also, the spatial variation in defocus blur differs much from that in motion blur, *e.g.*, transparency effects of moving objects in motion blur do not exist for out-of-focus objects in defocus blur. As a result, it is sub-optimal to directly call a motion deblurring method for SIDD.

Another straightforward implementation of introducing deep learning to SIDD is to replace the defocus map estimator in a traditional two-stage approach by a DNN-based method (*e.g.* [6]). Such an implementation still suffers from the issues existing in traditional methods, *i.e.* inaccurate estimation of $\boldsymbol{B}$ from a non-perfect defocus map and high computational cost for deblurring with a spatially-variant blurring operator. To fully exploit the potential of deep learning for SIDD, one needs to specifically design an end-to-end DNN that directly predicts the all-in-focus image from the defocused one. Recently, Abuolaim and Brown [18] developed an end-to-end DNN for constructing an all-in-focus image from a pair of images containing two sub-aperture views of the same scene. They also adapted their DNN to SIDD, but saw a significant performance decrease.

## 1.2 Main Idea

This paper aims at developing an end-to-end DNN for SIDD with better performance than existing methods, which is based on the following two derivations.

**GKM-based model for defocus blurring**   Since defocus PSFs show strong isotropy and smoothness, we propose to model the kernels $\{\boldsymbol{b}_{m,n}\}_{m,n}$ by *Gaussian kernel mixture (GKM)*:

$$\boldsymbol{b}_{m,n} = \sum_{k=1}^{K} \boldsymbol{\beta}_k[m,n]\boldsymbol{g}(\sigma_k), \tag{3}$$

where $\boldsymbol{g}(\sigma)$ denotes the 2D Gaussian kernel of variance $\sigma^2$, and $\boldsymbol{\beta}_k$ denotes the matrix of mixing coefficients for the $k$-th Gaussian kernel in the GKM. As the GKM can fit well most isotropic kernels, Eq. (3) is a more accurate model for real defocus PSFs than the often-used single Gaussian/disk form; see supplementary materials for a demonstration.

**Remark 1.** *The GKM degenerates to the single Gaussian form when only one mixing coefficient is* 1 *and the others are* 0 *for every location* $[m,n]$. *In general cases, the weighted summation of Gaussian kernels can represent non-Gaussian kernels, and thus the GKM can express a wider family of defocus PSFs than the single Gaussian form. There is also another work [9] that models defocus PSFs beyond single Gaussian kernels. It uses the generalized Gaussian function [9] where the parameters to be estimated are wrapped in a complex nonlinear function. In comparison, our GKM model is linear with pre-defined* $\{\sigma_k\}_k$, *which facilitates the estimation on its parameter* $\{\boldsymbol{\beta}_k\}_k$.

We can rewrite (1) as

$$\boldsymbol{y} = \sum_{m,n} \boldsymbol{\delta}_{m,n} \odot (\boldsymbol{b}_{m,n} \otimes \boldsymbol{x}) = \sum_{m,n}\sum_{k=1}^{K} \boldsymbol{\delta}_{m,n} \odot ((\boldsymbol{\beta}_k[m,n]\boldsymbol{g}(\sigma_k)) \otimes \boldsymbol{x}) = \sum_{k=1}^{K} \boldsymbol{\beta}_k \odot (\boldsymbol{g}(\sigma_k) \otimes \boldsymbol{x}),$$

where $\boldsymbol{\delta}_{m,n}$ denotes the Dirac delta centered at location $[m, n]$, and $\otimes, \odot$ denote the operations of 2D convolution and entry-wise multiplication, respectively. Then we have the GKM-based model for defocus blurring:

$$\boldsymbol{B} : \boldsymbol{x} \rightarrow \sum_{k=1}^{K} \boldsymbol{\beta}_k \odot (\boldsymbol{g}(\sigma_k) \otimes \boldsymbol{x}). \tag{4}$$

**Fixed-point iteration unrolling**  Recall that the blurring operator $\boldsymbol{B}$ is about keeping the low-frequency components and attenuating high-frequency ones of an image. Let $\boldsymbol{I}$ denote the identity mapping. The mapping $\boldsymbol{I} - \boldsymbol{B}$ is then about attenuating the low-frequency components and keeping the high-frequency ones. Neglecting the noise $\boldsymbol{\epsilon}$ and rewriting (1) by

$$\boldsymbol{x} = \boldsymbol{y} + (\boldsymbol{I} - \boldsymbol{B}) \circ \boldsymbol{x}, \tag{5}$$

we have then a fixed-point iteration for solving defocus deblurring, which is given by

$$\boldsymbol{x}^{(t+1)} = f(\boldsymbol{x}^{(t)}) = \boldsymbol{y} + \boldsymbol{x}^{(t)} - \boldsymbol{B} \circ \boldsymbol{x}^{(t)} = \boldsymbol{y} + \boldsymbol{x}^{(t)} - \sum_{k=1}^{K} \boldsymbol{\beta}_k \odot (\boldsymbol{g}(\sigma_k) \otimes \boldsymbol{x}^{(t)}), \text{ for } t = 1, 2, \dots . \tag{6}$$

Note that the fixed-point iteration above will be convergent if $\boldsymbol{I} - \boldsymbol{B}$ is a contractive mapping, or equivalently the eigenvalues of $\boldsymbol{B}$ fall in $(0, 1)$, which holds true when the defocus blurring is uniform with a normalized Gaussian kernel, *i.e.*, the scene depths are constant in the view.

Define $\sigma_1 = 0$ and $\boldsymbol{g}(\sigma_1) = \boldsymbol{\delta}$, so that clear regions can be modeled by setting $\boldsymbol{\beta}_1 = \mathbf{1}$ and zeroing $\boldsymbol{\beta}_k$ for $k > 1$. Let $\boldsymbol{\gamma}_1 = 1 - \boldsymbol{\beta}_1$ and $\boldsymbol{\gamma}_k = -\boldsymbol{\beta}_k$ for $k > 1$. The iteration (6) can be expressed as

$$\boldsymbol{x}^{(t+1)} = \boldsymbol{y}^{(t)} + \sum_{k=1}^{K} \boldsymbol{\gamma}_k \odot (\boldsymbol{g}(\sigma_k) \otimes \boldsymbol{x}^{(t)}), \text{ for } t = 1, 2, \dots . \tag{7}$$

In short, based on the GKM model of defocus PSFs, we can unroll a fixed-point iteration to solve (1) with learnable coefficient matrices $\boldsymbol{\gamma}_1, \cdots, \boldsymbol{\gamma}_K$. The motivation of unrolling a fixed-point iteration, instead of other iterative schemes such as gradient descent [19] and half quadratic splitting (HQS) [20], is to involve the forward operator $\boldsymbol{B}$ only, without introducing the transpose $\boldsymbol{B}^\top$ and the pseudo-inverse $\boldsymbol{B}^\dagger$.

**Remark 2.** *Our approach is sort of in the category of optimization unrolling, a widely-used methodology of designing DNNs for solving inverse problems. The key is to choose an appropriate iteration scheme that fits the problem well. Most existing optimization unrolling based image deblurring methods (e.g. [22, 23, 20, 24, 25, 26, 21, 27]) consider uniform blurring, where the matrix $\boldsymbol{B}$ can be represented by a convolution. The iterative schemes they adopt such as HQS, usually involve an inversion process for $\boldsymbol{B}$, which can be efficiently computed using FFT when $\boldsymbol{B}$ is a convolution operator. In our case, $\boldsymbol{B}$ is a spatially-varying blurring operator which does not have a computationally efficient inversion process. The proposed fixed-point iteration unrolling enables us to avoid such an inversion process in the DNN and use the forward operator only.*

The matrices $\boldsymbol{\gamma}_1, \cdots, \boldsymbol{\gamma}_K$ of mixing coefficients can be intezpreted as the attention maps associated to the feature maps generated by different Gaussian kernels. Thus, we construct a DNN with attention modules and long skip connections to utilize (7) for SIDD. In addition, we take a multi-scale scheme to implement the unrolling: at each iteration the DNN predicts the all-in-focus image at current scale and up-samples it for the calculation of the next iteration, with weight sharing used across scales. This leads to a scale-recurrent attentive DNN with a lightweight implementation.

## 1.3  Main Contributions

In comparison to existing two-stage or dual-view-based methods, this paper is among the first ones to present an end-to-end DNN for SIDD. See below for the summary of our technical contributions:

- A new and efficient parametric model based on GKM for defocus blur kernels, which fits real-world data better than existing models and thus leads to better performance in SIDD;

- A new formulation of the deblurring process derived from a fixed-point iteration so as to have a simple and efficient parameterization of defocus deblurring, which inspires an effective DNN for SIDD with low model complexity and high computational efficiency;

- A scale-recurrent attention mechanism which combines the coarse-to-fine progressive estimation and the unrolled deblurring process for better performance.

The experiments show that the proposed DNN brings noticeable improvement over existing approaches to SIDD, in terms of recovery quality, model complexity and computational efficiency.

## 2   Related Work

**Two-stage SIDD**   Most studies of two-stage methods for SIDD are concentrated on the first stage, *i.e.* defocus map estimation, while the second stage is often done by calling existing non-blind deconvolution methods (*e.g.* [28, 29, 11]). Defocus map estimation itself is a challenging task. There are several non-learning-based methods [2, 3, 5, 8] available for sparse defocus map estimation on edge points or regions. The dense defocus map is then constructed via some propagation method (*e.g.* Poisson matting [30]). Recently, deep learning has been extensively studied for defocus map estimation; see *e.g.* [7, 4, 6, 31]. As discussed in Section 1.1, the two-stage methods suffer from the sensitivity to estimation errors and high computational costs. In comparison, the proposed end-to-end DNN does not involve the defocus map estimation and thus does not suffer from these issues.

**End-to-end learning for defocus deblurring**   There are few works on learning an end-to-end DNN for defocus deblurring. Abuolaim and Brown [18] proposed to train an end-to-end U-Net for predicting an all-in-focus image from two view images captured by a dual-pixel sensor, and contributed a dataset of quadruples: a defocus blurred image, its all-in-focus counterpart, and two dual-pixel view images. Such a dual-pixel-based DNN showed impressive performance. However, its performance significantly decreases when being used for SIDD where only a single image is available for input. The prerequisite on dual-view inputs also limits the wider applicability of this method. In contrast, the proposed DNN is grounded by the defocus blurring model and only assumes a single image as input, thus being applicable to commodity cameras. Very recently, one parallel work to ours on end-to-end learning of SIDD was done by Lee *et al.* [32]. They proposed a DNN that predicts the pixel-wise filters for deblurring the deep defocused features of an image. In comparison, our DNN predicts the blurring filters for deblurring the image in an unrolled fixed-point iteration framework.

**DNNs for spatially-varying motion deblurring**   There have been many studies on deep learning for spatially-variant motion deblurring, especially on dynamic scenes with moving objects; see *e.g.* [12, 14, 13, 16, 15, 33, 34, 35, 36, 37, 17]. The methods using optical flow (*e.g.* [36]) or temporal cues of moving objects (*e.g.* [37]) for training are not applicable to SIDD. Many of these methods also adopt multi-scale structures (*e.g.* [13]) or attention mechanisms (*e.g.* [17]), but with generic designs which cannot effectively exploit the inherent characteristics of defocus blurring, *e.g.* strong isotropy and high correlation of the shapes of pixel-wise defocus PSFs. In comparison, our DNN is specifically designed for defocus deblurring and thus enjoys better performance in SIDD.

**Unrolling-based deep learning for image deblurring**   Unrolling-based DNNs have been extensively studied for non-blind image deblurring where the PSF is given as an input; see *e.g.* [20, 24, 21]. There are also some works [22, 23, 25, 27] on unrolling-based blind image deblurring where the PSF is unknown. However, these methods are restricted to the case of uniform blurring. The proposed unrolling-based DNN is the first one that can handle spatially-varying defocus blurring without given the PSFs, thanks to the proposed GKM-based defocus blurring model.

## 3   Network Architecture

The DNN we construct for SIDD, named as GKMNet (Gaussian Kernel Mixture Network), is based on the fix-point iteration (7) with a multi-scale recurrent fashion. See Figure 1 for the outline.

Given an input image $\boldsymbol{y}$, we generate its multi-scale versions $\boldsymbol{y}_1, \cdots, \boldsymbol{y}_T$ via bi-linear downsampling with factors $2^{T-1}, \cdots, 2^0$, respectively. Let $\boldsymbol{x}_1 = \boldsymbol{y}_1$ and let $\uparrow_2$ denote the upsampling operation by factor 2. We consider the following multi-scale extension of (7):

$$\boldsymbol{x}_{t+1} = \boldsymbol{y}_t + \sum_{k=1}^{K} \boldsymbol{\gamma}_{t,k} \odot (\boldsymbol{g}(\sigma_k) \otimes \boldsymbol{x}_t \uparrow_2), \text{ for } t = 1, \cdots, T, \tag{8}$$

The GKMNet employs $T$ recurrent blocks to implement (8). The $t$-th block takes $\boldsymbol{y}_t$ and $\boldsymbol{x}_t{\uparrow_2}$ as input and outputs $\boldsymbol{x}_{t+1}$. There are two modules in each block: a Gaussian convolution module (GCM) and a scale-recurrent attention module (SRAM). The GCM performs the Gaussian filtering in (8) to provide $K$ feature maps, denoted by $\{\boldsymbol{z}_{t,1}, \ldots, \boldsymbol{z}_{t,K}\}$, for the $t$-th block. The SRAM generates the corresponding coefficient maps $\{\boldsymbol{\gamma}_{t,1}, \ldots, \boldsymbol{\gamma}_{t,K}\}$ from $\boldsymbol{y}_t$. The output of the $t$-th block is given by

$$\boldsymbol{x}_t = \boldsymbol{y}_t + \sum_{k=1}^{K} \omega_{t,k}(\boldsymbol{\gamma}_{t,k} \odot \boldsymbol{z}_{t,k}), \tag{9}$$

where the weights $\{\omega_{t,k}\}_{t,k}$ are for scaling the mixing coefficients $\{\boldsymbol{\gamma}_{t,k}\}_{t,k}$ predicted by the SRAM within a certain range. The weighted summation with $\{\omega_{t,k}\}_{t,k}$ is implemented by $1 \times 1$ convolution. The output of the $T$-th block, *i.e.* $\boldsymbol{x}_T$, is used as the final deblurring result. Note that the GKMNet relates the scales not only by passing the output from one scale to the next, but also by the recurrence mechanism built in the SRAM. Let $\boldsymbol{x}_t^{\text{gt}}$ denotes the downsampled ground truth of the same size as $\boldsymbol{x}_t$. The training loss is defined by the supervision at all scales:

$$\mathcal{L} := \sum_{t=1}^{T} \mathcal{C}(\boldsymbol{x}_t, \boldsymbol{x}_t^{\text{gt}}), \text{ for some cost function } \mathcal{C}(\cdot, \cdot), \tag{10}$$

where we assign equal weights to the losses at different scales for simplicity.

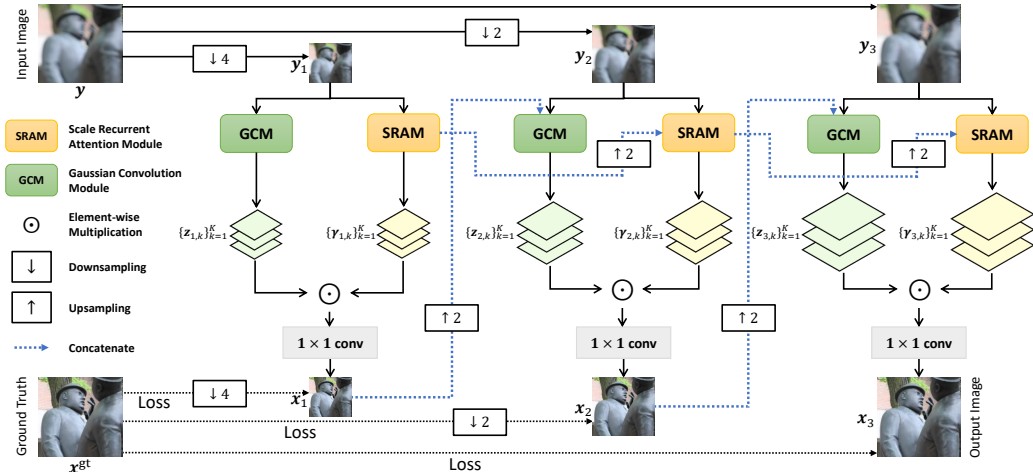

Figure 1: Diagram of proposed GKMNet for SIDD with $T = 3$.

**Gaussian Convolution Module** The GCM is a simple group convolutional layer defined by a series of pre-designed 2D Gaussian kernels applied to the R, G, B channels, respectively. The kernel sizes are set to $1 \times 1, 3 \times 3, 5 \times 5, \cdots, M \times M$. For a kernel size $m \times m$ with $m > 1$, we generate two Gaussian kernels with $\sigma$ set to $\frac{1}{4}(m-2)$ and $\frac{1}{4}(m-1)$ respectively. The Gaussian kernel of size $1 \times 1$ is the Dirac delta kernel. As a result, there are totally $M$ different kernels. Their parameters are fixed across all scales. The convolution kernels in GCM are not learned for two reasons: *(i)* it reduces the model complexity and leads to faster training; and *(ii)* the learned convolution kernels do not benefit the performance as empirically observed. Note that in addition to $\boldsymbol{x}_t$, we also input $\boldsymbol{y}_t$ in parallel for feeding more information to the next stage.

**Scale-Recurrent Attention Module** The SRAM maps an input image $\boldsymbol{y}_t$ to the coefficient maps $\{\boldsymbol{\gamma}_{t,k}\}_{k=1}^{K}$. The coefficient maps can be viewed as the spatial-channel attention maps associated to the feature maps generated by different Gaussian kernels. We thus draw inspirations from existing attention modules (*e.g.* [13, 41, 17]) to have a lightweight design on SRAM. As illustrated in Figure 2, the SRAM consists of *(i)* an attentive encoder-decoder backbone [38] for feature extraction; and *(ii)* an attention prediction unit (APU) based on Conv-LSTM (convolutional long short-term memory) [39].

The attentive encoder-decoder backbone sequentially connects a convolutional layer, two encoder blocks, and two decoder blocks. Each encoder/decoder block contains one convolutional layer with

downsampling/upsampling, two convolutional layers with residual connections, and a triplet attention block [40]. The triplet attention is used to improve the NN's spatial and channel adaptivity for better prediction, which can be viewed as an attention-in-attention mechanism in the SRAM. Given a feature tensor $\mathcal{X} \in \mathbb{R}^{C \times H \times W}$ defined in the channel-height-width space, the triplet attention block generates three parallel attention maps $\boldsymbol{a}_{\mathrm{W}} \in \mathbb{R}^{C \times H}$, $\boldsymbol{a}_{\mathrm{H}} \in \mathbb{R}^{C \times W}$, $\boldsymbol{a}_{\mathrm{C}} \in \mathbb{R}^{H \times W}$ from the channel-height, channel-width and height-width slices of $\mathcal{X}$ respectively, and then applies them for re-calibrating $\mathcal{X}$ so as to encode spatial-channel dependencies into the features.

The APU contains two paths. The first path mainly contains a Conv-LSTM and applies downsampling/upsampling before/after it for inducing local smoothness on the predicted coefficient maps. The second path contains two convolutional layers, but without downsampling/upsampling for preserving detailed information for the prediction. Let $\bar{\boldsymbol{\gamma}}_1, \bar{\boldsymbol{\gamma}}_2$ denote the predictions from these two paths. Then the final coefficient map is predicted by $\boldsymbol{\gamma} = \tanh(\bar{\boldsymbol{\gamma}}_1 \odot \bar{\boldsymbol{\gamma}}_2 + \bar{\boldsymbol{\gamma}}_1)$. Such a design is motivated from the Squeeze-and-Attention [41]. The Conv-LSTM at the first path in the APU is used for capturing the dependencies among the blur amount at different scales. The hidden states in the Conv-LSTM can capture useful information from different scales and benefit the restoration across scales. This enables the SRAM to progressively improve the estimation on the coefficient maps.

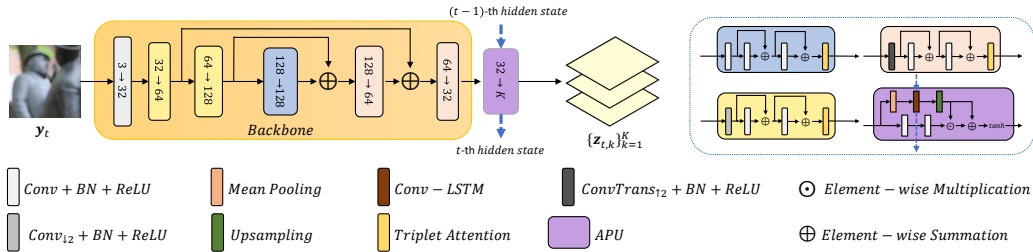

Figure 2: Diagram of SRAM. All convolutional kernels are of size $3 \times 3$.

**Remark 3.** *Unlike many existing unrolling-based DNNs (e.g. [23, 20, 24, 25, 26, 21, 27]) for image deblurring, the GKMNet does not have any explicit artifact removal block. However, it still performs well empirically, as shown in the experiments. The reasons may be as follows. First, the inversion process implemented by the GCM and SRAM is not the classic non-learnable estimator in existing unrolling-based DNNs, but the one containing learnable blocks. Thus, the GKMNet can be viewed as some forms of the inversion process with implicit built-in artifact removal. For instance, in the GKM model, the learned combination of Gaussian kernels not only can express the defocus kernels, but also may encode a component to suppress the artifacts. Second, there is probably a regularization effect due to the fact that the fixed point iterations do not necessarily converge to a fixed point (similar to early stopping), as well as a regularization effect due to the relaxation of the fixed-point iteration to the multi-scale iteration. While the exact nature of the regularizations and built-in artifact removal is unknown, it does not necessarily prevent the GKMNet from working, since the parameters are end-to-end learned in a supervised manner.*

## 4 Experiments

There are few datasets available for benchmarking defocus deblurring. The most well-known one is the DPD dataset [18]. It provides 500 pairs of images with defocus blur and their corresponding all-in-focus images, as well as the two associated sub-aperture views called dual-pixel images, all in 16-bit color. The training/validation/test splits in the dataset consist of 350/74/76 samples. Our GKMNet is trained and tested on the DPD dataset without using the dual-pixel images. In addition, we test GKMNet on the RTF test set [7] as well as the images from CUHK-BD dataset [42].

Three image quality metrics are used for quantitative evaluation, including two standard metrics: PSNR (Peak Signal to Noise Ratio) and SSIM (Structural Similarity Index Measure) [43], and the LPIPS (Learned Perceptual Image Patch Similarity) [44] for perceptual quality (also used in [18]). The GKMNet is also compared to other DNNs in terms of model complexity and computational efficiency. The model complexity is measured by three metrics: number of parameters, number of FLOPs (floating-point operations per second), and model size. The computational efficiency is measured by the average inference time on an image of $1680 \times 1120$ pixels, tested on an Intel

i5-9600KF CPU and on an NVIDIA GTX 1080Ti GPU, respectively. For non-DNN-based methods, only the inference time on CPU is reported.

Through all experiments, the maximum size of Gaussian kernels in GCM is set to $M = 21$. The number of scales is set to $T = 3$. The learnable parameters in SRAM are initialized by Xavier [45]. The Adam optimizer [46] is used for training with 3000 epochs and batch size 4. The learning rate is fixed at $10^{-4}$ in the first 2000 epochs and decayed to $10^{-5}$ in the last 1000 epochs. The cost function in (10) is set to the squared $\ell_2$ loss in the first 1000 iterations, and alternatively set to the SSIM loss and squared $\ell_2$ loss every 500 epochs afterwards. Data augmentation is done by random cropping to $256 \times 256$ pixels. The code of GKMNet is available at `https://github.com/csZcWu/GKMNet`.

### 4.1 Evaluation on DPD Dataset

The methods for comparison include JNB [2], EBDB [5], DMENet [6] and DPDNet [18]. The DPDNet [18] has two versions: DPDNet-D taking two dual-pixel images as input and DPDNet-S only taking a single image as input. Both versions are included for comparison whose results are reproduced by the pre-trained models from their authors. The JNB, EDBD and DMENet are all two-stage methods focusing on defocus map estimation, with results quoted from [18]. In addition, we also include two DNNs of dynamic scene deblurring: SRN [13] and AttNet [17]. These two DNNs also adopt a multi-scale architecture. Both DNNs are retrained using the same data as ours.

See Table 1 for the quantitative comparison on deblurring performance. Our GKMNet outperforms all other compared methods. The performance of EBDB, DMENet and JNB is much worse than that of GKMNet. This is probably because the errors in their defocus maps are magnified in the consequent deconvolution process. It indicates the advantage of the end-to-end learning of GKMNet which does not require defocus map estimation. The GKMNet also outperforms DPDNet-S by a large margin. This is not surprising as DPDNet-S is originally designed for defocus deblurring on two view images, not a single one. Also unsurprisingly, GKMNet outperforms SRN and AttNet, two DNNs designed for dynamic scene deblurring rather than SIDD. Surprisingly, our GKMNet with a single-image input even outperforms DPDNet-D which takes dual-pixel images as input. All these results have clearly demonstrated the effectiveness of the proposed GKMNet on SIDD. See supplementary materials for the visualization of coefficient maps generated by the SRAM.

Table 1 also lists the results on model complexity and computational efficiency. In terms of all three metrics, GKMNet's complexity is much lower than that of other deep models. For instance, around $1/23$ of DPDNet and $1/7$ of SRN for number of parameters, and around $1/62$ of DPDNet for model size. Such low complexity comes from the compact architecture of GKMNet. Regarding the running time, GKMNet is also much faster than the JNB, EBDB and DEMNet, as these methods need to call an iterative deconvolution post-process which is slow. GKMNet is about ten time faster than DPDNet, and its speed is comparable to SRN and AttNet. It is noted that GKMNet has a much less training time due to its low complexity. Its training on the DPD dataset takes less than 32 hours, while SRN and AttNet take nearly three days, on an NVDIA GTX 1080Ti GPU.

Table 1: Quantitative comparison of different methods on DPD test set.

| Model | PSNR (dB) | SSIM | LPIPS | #Parameters (Million) | #FLOPs (Billion) | Model Size (MegaBytes) | Time (Seconds) CPU | GPU |
|---|---|---|---|---|---|---|---|---|
| JNB [2] | 23.84 | 0.715 | 0.315 | - | - | - | 843.1 | - |
| EBDB [5] | 23.45 | 0.683 | 0.336 | - | - | - | 929.7 | - |
| DMENet [6] | 23.41 | 0.714 | 0.349 | 26.94 | - | - | 613.7 | - |
| DPDNet-S [18] | 24.34 | 0.747 | 0.277 | 32.25 | 4042.5 | 355.3 | 56.6 | 0.41 |
| DPDNet-D [18] | 25.12 | 0.786 | 0.223 | 32.25 | 4048.6 | 355.3 | 56.6 | 0.41 |
| SRN [13] | 24.61 | 0.612 | 0.265 | 10.25 | 3117.6 | 32.0 | 58.3 | **0.032** |
| AttNet [17] | 25.22 | 0.781 | 0.219 | 6.91 | 3450.4 | 39.5 | 62.8 | 0.041 |
| GKMNet [Ours] | **25.47** | **0.789** | **0.219** | **1.41** | **603.5** | **5.7** | **43.9** | 0.040 |

As mentioned in Remark 3, without any explicit artifact/noise removal block in the GKMNet, the robustness of GKMNet is one concern. Thus, we evaluate the robustness to image noise using the DPD dataset with additional corruption by additive Gaussian white noise with standard deviation $\hat{\sigma} = 1, 3, 5$ respectively. The DPDNet-D, SRN and AttNet are selected for comparison. Both the

models trained on the original DPD training set and those retrained on the noisier DPD training set are included. The later ones are marked with a "+". See Table 2 for the results. In the presence of additional noise, our GKMNet still outperforms other models for both versions of training data.

Table 2: Quantitative comparison of different methods on noisier DPD test set.

| Noise $\hat{\sigma}$ | DPDNet-D | | | SRN | | | AttNet | | | GKMNet | | |
|---|---|---|---|---|---|---|---|---|---|---|---|---|
| | PSNR | SSIM | LPIPS | PSNR | SSIM | LPIPS | PSNR | SSIM | LPIPS | PSNR | SSIM | LPIPS |
| 1 | 25.08 | 0.788 | 0.220 | 24.58 | 0.692 | 0.315 | 25.18 | 0.783 | 0.221 | **25.43** | **0.784** | **0.225** |
| 3 | 24.73 | 0.746 | 0.248 | 24.48 | 0.668 | 0.366 | 24.77 | 0.749 | 0.261 | **25.02** | **0.758** | **0.246** |
| 5 | 24.07 | 0.665 | 0.441 | 24.30 | 0.628 | 0.411 | 24.36 | 0.660 | 0.347 | **24.61** | **0.712** | **0.299** |
| Noise $\hat{\sigma}$ | DPDNet-D+ | | | SRN+ | | | AttNet+ | | | GKMNet+ | | |
| | PSNR | SSIM | LPIPS | PSNR | SSIM | LPIPS | PSNR | SSIM | LPIPS | PSNR | SSIM | LPIPS |
| 1 | 25.09 | 0.788 | 0.220 | 24.61 | 0.697 | 0.316 | 25.18 | 0.783 | 0.221 | **25.48** | **0.789** | **0.221** |
| 3 | 24.76 | 0.748 | 0.248 | 24.53 | 0.672 | 0.360 | 24.85 | 0.753 | 0.255 | **25.12** | **0.760** | **0.247** |
| 5 | 24.37 | 0.710 | 0.379 | 24.34 | 0.632 | 0.399 | 24.48 | 0.681 | 0.332 | **24.92** | **0.742** | **0.273** |

## 4.2 Evaluation on RTF Dataset and CUHK-BD Sample Images

The RTF dataset [7] contains 22 pairs of defocused and all-in-focus images. Following [7], in addition to the original images, two noisier versions are generated by adding Gaussian white noise with standard deviation $\hat{\sigma} = 1$ and $2.55$ respectively. The DPDNet-S, SRN and AttNet are selected for comparison. Note that the training set of RTF is not available. Thus, we directly apply the models that are respectively trained on the original DPD dataset and the noisier DPD dataset. See Table 3 for the results. The GKMNet again outperforms other DNNs. Since the models art trained and tested with different datasets, such results also demonstrate the generalizability of our GKMNet.

Table 3: Quantitative comparison of different methods on RTF test set.

| Noise $\hat{\sigma}$ | DPDNet-S | | | SRN | | | AttNet | | | GKMNet | | |
|---|---|---|---|---|---|---|---|---|---|---|---|---|
| | PSNR | SSIM | LPIPS | PSNR | SSIM | LPIPS | PSNR | SSIM | LPIPS | PSNR | SSIM | LPIPS |
| 0 | 23.61 | 0.597 | 0.296 | 23.71 | 0.617 | 0.324 | 25.45 | 0.802 | 0.219 | **25.72** | **0.811** | **0.211** |
| 1 | 23.58 | 0.592 | 0.332 | 23.68 | 0.611 | 0.311 | 24.99 | 0.797 | 0.243 | **25.56** | **0.798** | **0.207** |
| 2.55 | 23.44 | 0.576 | 0.344 | 23.50 | 0.591 | 0.276 | 23.83 | 0.727 | 0.257 | **25.01** | **0.761** | **0.205** |
| Noise $\hat{\sigma}$ | DPDNet-S+ | | | SRN+ | | | AttNet+ | | | GKMNet+ | | |
| | PSNR | SSIM | LPIPS | PSNR | SSIM | LPIPS | PSNR | SSIM | LPIPS | PSNR | SSIM | LPIPS |
| 0 | 23.65 | 0.605 | 0.290 | 23.70 | 0.617 | 0.325 | 25.40 | 0.792 | 0.225 | **25.69** | **0.808** | **0.212** |
| 1 | 23.59 | 0.594 | 0.329 | 23.65 | 0.605 | 0.313 | 25.06 | 0.800 | 0.236 | **25.60** | **0.802** | **0.206** |
| 2.55 | 23.49 | 0.581 | 0.341 | 23.54 | 0.597 | 0.281 | 24.33 | 0.749 | 0.250 | **25.21** | **0.787** | **0.210** |

The CUHK-BD dataset [42] contains $704$ defocused images without ground truths. We select some of its images for test. See Figure 3 for the visual inspection on the results. Our GKMNet can successfully restore fine details with less visual artifacts, in comparison to DPDNet-S, SRN and AttNet. More visual comparison can be found in supplementary materials. It is worth mentioning that the defocus PSFs of the image synthesized from dual-pixel images are slightly different from the ones in real images [47, 48, 49]. However, our GKMNet still generalizes well.

## 4.3 Ablation Study

To evaluate the performance contribution of each component in GKMNet, we implement the following baselines from GKMNet for comparison, which are trained in the same way as the original GKMNet. (a) GKMNet(1): Use only the original image scale in GKMNet with $T = 1$. (b) w/o Conv-LSTM: Replace the Conv-LSTM block at the APU by a convolutional layer with the same kernel size, and share its weights across scales for relating different scales; (c) SRAM*: Use the SRAM as a whole DNN for SIDD, with an $1 \times 1$ convolutional layer with Sigmoid activation attached for outputting an image; (d) Learned GCM (GKM): The kernels in GCM are initialized by the predefined Gaussian kernels and learned with the DNN. (e) Learned GCM (Rand): The kernels in GCM are randomly

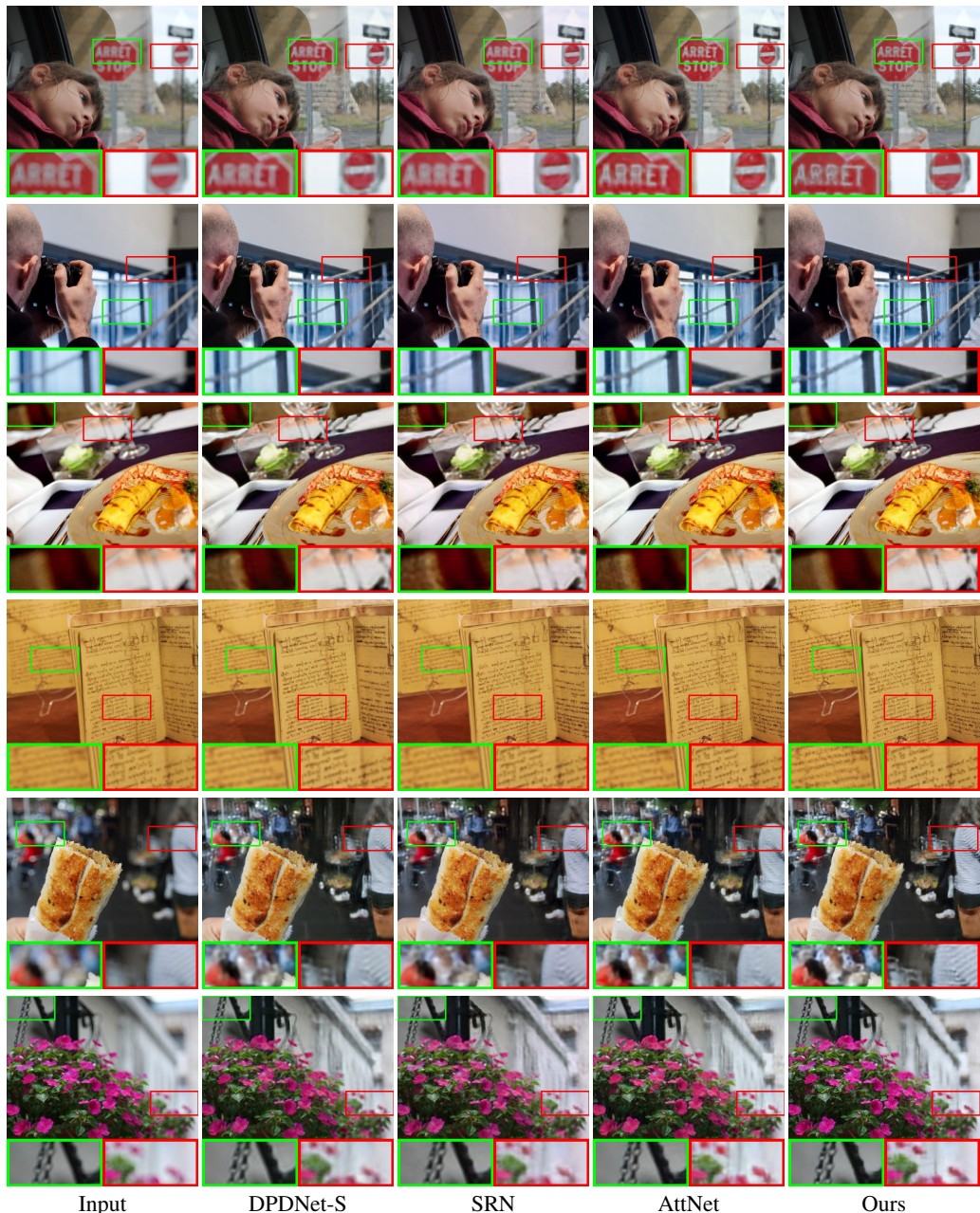

| Input | DPDNet-S | SRN | AttNet | Ours |

Figure 3: Visual comparison of SIDD results of different methods on CUHK-BD sample images.

initialized by Xavier [45] and learned with the DNN. Note that weight sharing is adopted for GCM across scales in (d) and (e) because it leads to performance improvement.

See Table 4 for the results of the baselines, from which we have the following observations. (a) The multi-scale estimation scheme in SRAM leads to significant improvement over the single-scale one, with only a few additional model parameters introduced. (b) The improvement from the Conv-LSTM over simple cross-scale weight-sharing convolutions is significant, which implies that the Conv-LSTM can effectively exploit the features learned from different scales to guide the deblurring process. (c) Directly using the SRAM as an end-to-end DNN for SIDD leads to much worse results, which justifies the effectiveness of our specific DNN design. (d) Learning the GCM with Gaussian kernel initialization leads to a very minor improvement over using fixed Gaussian kernels in GCM. (e) Learning the GCM with a random initialization even yields slightly worse results. Both (d) and (e)

indicate the effectiveness of our GKM model for defocus PSFs. See also Figure 4 for the kernels w/o and w/ learning in GCM. The kernels learned with (d) are very close to original predefined ones.

Table 4: Quantitative comparison of GKMNet and its baselines.

| Model | PSNR (dB) | SSIM | LPIPS | #Parameters (Million) | #FLOPs (Billion) | Model Size (MegaBytes) | Time (Seconds) |
|---|---|---|---|---|---|---|---|
| SRAM* | 21.98 | 0.724 | 0.251 | 1.41 | 606.1 | 5.30 | 0.040 |
| ASGMNet(1) | 23.63 | 0.723 | 0.248 | 1.41 | 459.7 | 5.70 | **0.036** |
| w/o Conv-LSTM | 25.00 | 0.774 | 0.231 | **1.11** | **418.7** | **4.33** | 0.036 |
| Learned GCM (Rand) | 25.35 | 0.787 | 0.223 | 1.41 | 603.5 | 5.70 | 0.040 |
| Learned GCM (GKM) | **25.49** | **0.790** | **0.217** | 1.41 | 603.5 | 5.70 | 0.040 |
| GKMNet | 25.46 | 0.789 | 0.219 | 1.41 | 603.5 | 5.70 | 0.040 |

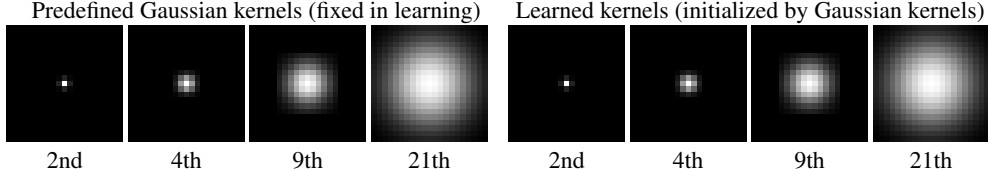

Predefined Gaussian kernels (fixed in learning)     Learned kernels (initialized by Gaussian kernels)

2nd     4th     9th     21th          2nd     4th     9th     21th

Figure 4: Visualization of predefined kernels and learned kernels in GCM.

Recall that the fixed point iteration (7) is applied to multiple iterations at the same scale, while our actual architecture also simultaneously increases the scale. We could further write our architecture as $S$ scales and $R$ iterations per scale, where we use $S = 3, R = 1$ in the GKMNet. To further verify the effectiveness of the multi-scale estimation in GKMNet, Table 5 lists the results using different values of $S$ and $R$. It shows that using single or fewer scales with multiple iterations noticeably decreases the performance and increases the inference time. Using an additional iteration with $S = 3$ only brings a minor improvement but nearly doubles the complexity and time. Thus, our coarse-to-fine estimation scheme is a better implementation for high performance and low computational cost.

Table 5: Quantitative comparison of extended GKMNet models using different values of $S$ and $R$.

| Model | PSNR (dB) | SSIM | LPIPS | Parameters (Million) | FLOPs (Billion) | Model Size (MegaBytes) | Time (Seconds) |
|---|---|---|---|---|---|---|---|
| S=1, R=3 | 25.11 | 0.773 | 0.237 | 1.41 | 1384.2 | 5.7 | 0.069 |
| S=2, R=3 | 25.13 | **0.789** | 0.205 | 4.22 | 1730.3 | 17.1 | 0.122 |
| S=3, R=2 | **25.53** | 0.773 | **0.201** | 2.81 | 1211.2 | 11.3 | 0.081 |
| S=3, R=1 | 25.47 | **0.789** | 0.219 | **1.41** | **603.5** | **5.7** | **0.04** |

## 5   Conclusion

Defocus blur often occurs in images and has its own characteristics from motion blur. This paper proposed a DNN for SIDD with strong motivations from unrolling a fixed-point iteration derived from a GKM-based model of defocus blurring process. Together with a scale-recurrent implementation, we developed a lightweight DNN with state-of-the-art performance. In future, we will study the extension to other image deblurring problems such as dynamic scene deblurring.

While our approach achieved state-of-the-art results on existing datasets, its implicit regularizations and built-in artifact removal mechanisms are not very clear. In addition, its performance on severely blurred regions is not that satisfactory, with much room for improvement. These issues will also be studied in our further work.

## Acknowledgments and Disclosure of Funding

Yuhui Quan would like to acknowledge the support from National Natural Science Foundation of China (Grant No. 61872151) and CCF-Tencent Open Fund 2020. Hui Ji would like to acknowledge

the support from Singapore MOE Academic Research Fund (AcRF) Tier 1 Research Grant (R-146-000-315-114).

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
