# Supplementary Materials

**Yuhui Quan,    Zicong Wu**
School of Computer Science and Engineering
South China University of Technology

**Hui Ji**
Department of Mathematics
National University of Singapore

## Contents

## 1  GKM model vs. Single Gaussian (Disk) Model

In this study, we evaluate the accuracy of the models for defocus PSFs: the proposed GKM-based model versus the often-used single Gaussian (disk)-based model. As there is no ground truth PSFs available in public domain, we indirectly evaluate the accuracy in terms of the synthesis quality of defocus images from these models.

Given a (clear, defocused) image pair on the RTF test set, we first generate the blurred versions of the clear image with the Gaussian kernels defined in the GCM. Then the coefficient maps are optimized to combine the blurred versions to approximate the given defocused image. After that, entry-wise multiplication and element-wise summation are done following the GKM-based defocus blurring model to synthesize a defocused image.

For comparison, we also generate defocus blurred images using the single Gaussian-based model and single disk-based model, respectively. Concretely, we use Gaussian (disk) kernels to obtain multiple blurred versions of the input image.[1] Then, to synthesize the defocused image, for each spatial location, we pick up the pixel from the blurred versions which have the closest Euclidean distance to its corresponding pixel in the given defocused image.

Table 1: Quantitative comparison of different models on defocused image synthesis.

| Model | Single Disk | Single Gaussian | GKM |
|---|---|---|---|
| PSNR(dB) | 32.07 | 33.35 | 54.39 |
| SSIM | 0.923 | 0.943 | 0.998 |

To evaluate the quality of the synthesized defocused images, we calculate the PSNR values and SSIM values between the synthesized images and the given defocused images. The average results are given in Table 1. It can be seen that our GKM-based model outperforms both the single Gaussian-based

---

[1]The same set of Gaussian kernels as that of the GKM model is used.

35th Conference on Neural Information Processing Systems (NeurIPS 2021).

model and the single disk-based model by a large margin in synthesizing defocus blur, in terms of the approximation to real-world defocused images. Such a noticeable improvement implies that the proposed GKM model enjoys higher accuracy than existing single Gaussian (disk) models for expressing real-world defocus PSFs. See also Figure 1 for a visual inspection on the synthetic results.

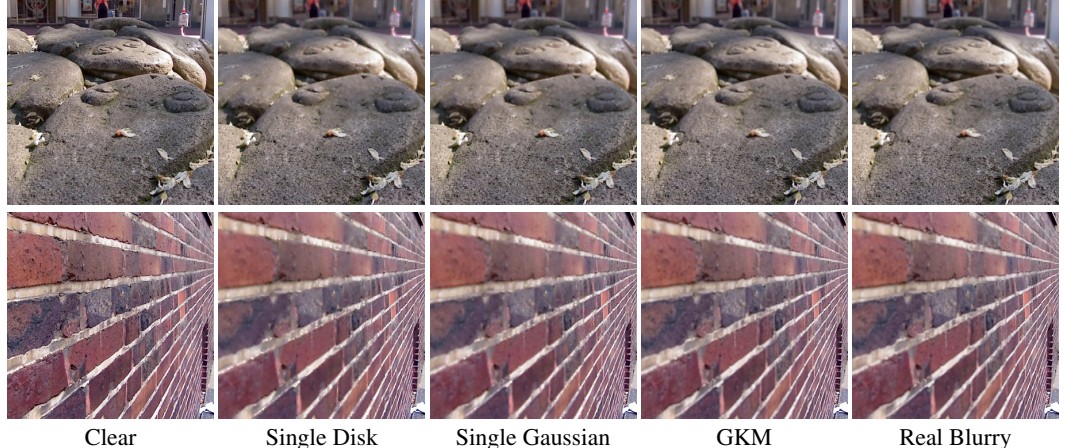

| Clear | Single Disk | Single Gaussian | GKM | Real Blurry |

Figure 1: Visual comparison of synthesized images of different models on RTF test set.

## 2    Comparison to HQS-based Unrolling

Most existing unrolling-based DNNs for image deblurring are for non-blind image deconvolution. In non-blind image deconvolution, the blurring is assumed to be uniform which can be modeled by the convolution of a single convolution kernel $k$, and it needs to solve a linear system $y = k \otimes x + \epsilon$. As the inversion of such a convolution can be efficiently implemented via FFT, existing unrolling methods (e.g. HQS, ADMM) often use FFT to run the inversion during each iteration and then call a DNN module to refine the inversion results.

The defocus deblurring problem is very different from non-blind image deconvolution. It belongs to the category of non-uniform blind deblurring, which cannot be modeled by a convolution and needs to estimate both the blurring operator $B$ and the clean image $x$ in $y = Bx + \epsilon$. Thus, the inversion in HQS-based or ADMM-based unrolling cannot be efficiently computed via FFT. Therefore, we proposed a fixed-point iteration scheme which does not involve the inversion of the operator $B$ but the forward operator $B$ itself only.

To demonstrate that, we evaluate the computational efficiency of an HQS-based variant constructed based on the popular HQS+CG unrolling-based non-blind deblurring framework, where the conjugate gradient (CG) solver is used for the inversion subproblem and a CNN-based denoiser block is used as the artifact removal block. We combine the blurring operator estimated by our SRAM+GCM into the HQS+CG framework, and compare its complexity with our model. For fair comparison, we adopt the same multi-scale estimation and cross-scale weight sharing schemes as our GKMNet, so as to reduce the complexity of the variant. The number of CG iterations in each scale, denoted by $P$, is set to 10,15,20,25, respectively, as such values are often used in existing CG-unrolling-based DNNs for image deblurring. See Table 2 for the comparison. The computational efficiency of such an HQS-based variant is noticeably lower than our GKMNet.

Table 2: Comparison on the effect of different numbers of recurrent blocks.

| Method | #Parameters (Million) | #FLOPs (Billion) | Time (Seconds) |
|---|---|---|---|
| HQS + CG ($P = 10$) | 3.26 | 4994.3 | 0.308 |
| HQS + CG ($P = 15$) | 3.26 | 7264.5 | 0.485 |
| HQS + CG ($P = 20$) | 3.26 | 9534.7 | 0.611 |
| HQS + CG ($P = 25$) | 3.26 | 11804.9 | 0.824 |
| GKMNet | **1.41** | **603.5** | **0.040** |

# 3  Visual Comparison of Results on Some Sample Images of DPD Dataset

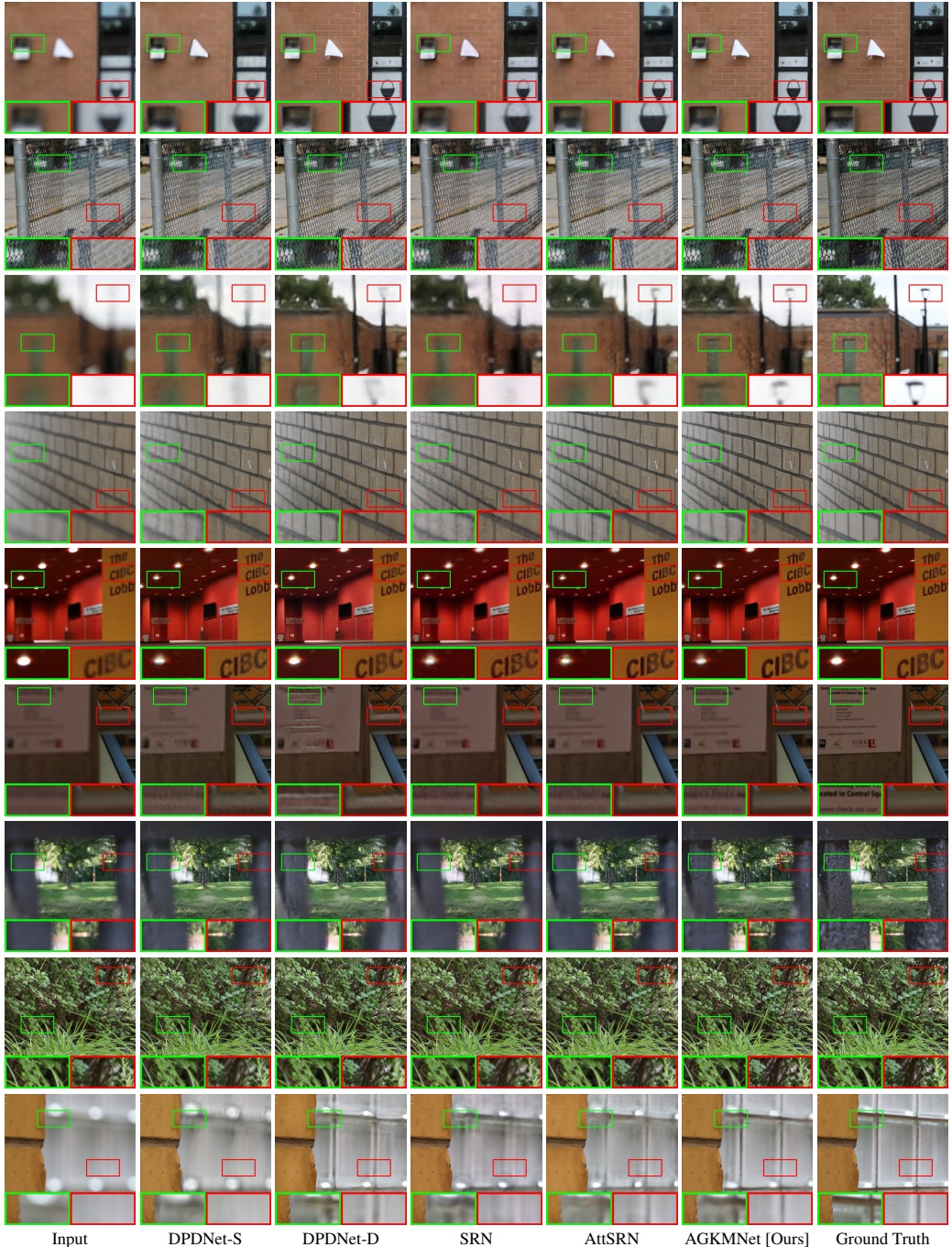

|       |          |          |     |        |                 |              |
| Input | DPDNet-S | DPDNet-D | SRN | AttSRN | AGKMNet [Ours]  | Ground Truth |

Figure 2: Visual comparison of deblurring results of different methods on sample images of DPD dataset.

# 4 Visual Comparison of Results on Some Sample Images of RTF Dataset

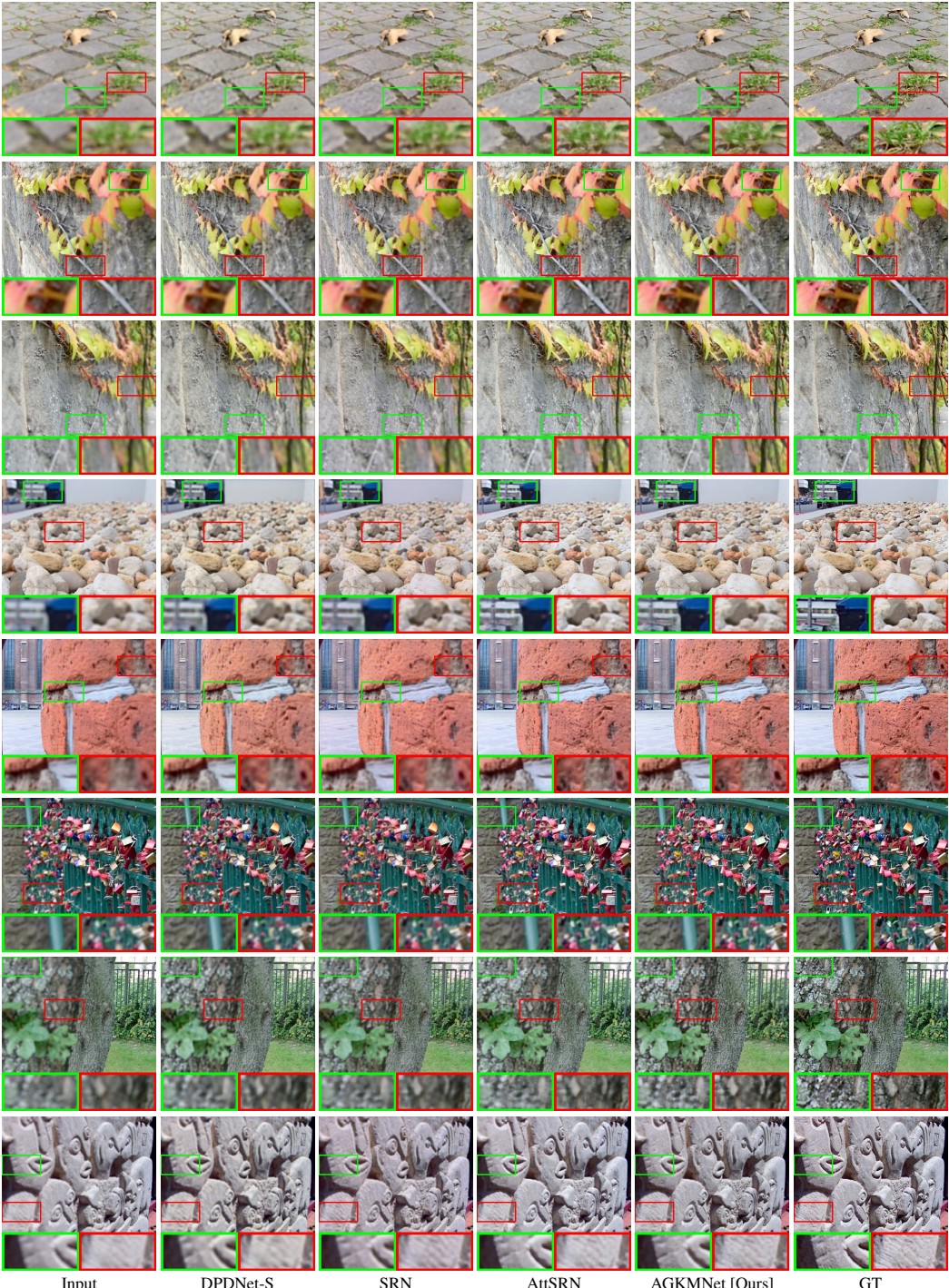

| Input | DPDNet-S | SRN | AttSRN | AGKMNet [Ours] | GT |

Figure 3: Visual comparison of deblurring results of different methods on sample images of RTF dataset with $\sigma = 0$.

# 5 Visualization of Coefficient Maps

Figure 4 visualizes the coefficient maps from the SRAM at the last scale, which correspond to the Gaussian kernels of sizes $1 \times 1$, $13 \times 13$, $15 \times 15$, $19 \times 19$, respectively.

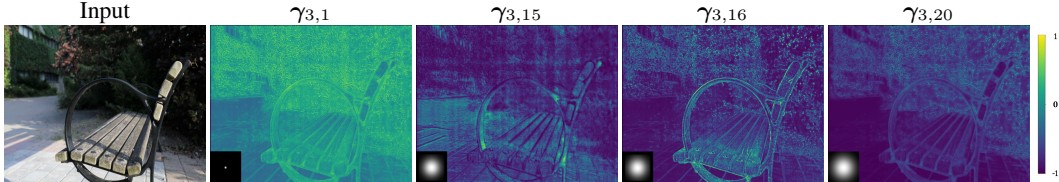

Figure 4: Visualization of coefficient maps associated to the 1st, 15th, 16th, 20th Gaussian kernels at the original image scale.