# OpenReview forum: "Gaussian Kernel Mixture Network for Single Image Defocus Deblurring"
_NeurIPS.cc/2021/Conference — NeurIPS 2021 Poster_

### Official Review · Reviewer_SpFs · 2021-07-14

**Rating:** 5
**Confidence:** 4

**Summary:**

This paper proposed a deep attentive network for defocus blur removal. The basic idea is the unfolding of a fixed point algorithm for defocus blur removal using a Gaussian Scale Mixture model for modeling the spatially varying blur kernels. The proposed network is a multi-scale architecture with GCM block and SRAM block. The proposed network is evaluated on several defocus blur datasets, and showed improved performance compared with some other methods, e.g., AttNet, ASG, etc.

**Limitations And Societal Impact:**

Removing defocus blur is a fundamental task in image processing, benefiting to applications such as auto-focus, depth from defocus, downstream recognition tasks, etc. The paper did not fully discuss on the limitations of the proposed approach.

**Main Review:**

The major interesting contributions of this work are the proposed GSM modeling of  defocus blur and the attentive module to estimate the component coefficients. The proposed network is inspired by unfolding a fixed point algorithm for deblur in Eqn. 7.

1. On the fixed point iterations.

The fixed point iterations in Eqn. (5) is not fully analyzed, for example, its convergence analysis. Because it seems that the proposed network is faithfully implementing this iterations in GCM block, these analysis are important to understand the network performance.

2. Whether the network learn to remove artifacts.

It is not new to unfold algorithms, such as HQS, for image deconvolution. What is the advantage of this proposed fixed point algorithm (and its derived network) compared with these previous HQS-based unfolded networks for image deblur? Moreover, it seems that the proposed network does not explicitly have the artifact removal block (commonly corresponds to proximal operator of regularizer). How to guarantee that the restored sharp image is artifact-free?

3. Comparisons.

It is true that there are very few papers focusing on the blind defocus blur removal. However, the proposed approach can be further compared with other possible variants, such as different unfolded network structures (e.g., HQS) .

Are the proposed network trained once, and tested on different datasets? Is the trained network generalizable to defocus image captured by different cameras?

**Time Spent Reviewing:**

2.5 hours

---

> ### Author Response · Authors · 2021-08-10
> **Responses to the comments from Reviewer SpFs**
>
> We sincerely thank the reviewer for his/her constructive comments.
>
> ------------------------------------------------------------------------------
> **Regarding the convergence of the iteration**
>
> For a fixed-point iteration, it will be convergent if the matrix $I-B$ is a contractive mapping, or equivalently the eigen-values of $B$ fall in (0,1) , which holds true if the blurring is a uniform one with a normalized Gaussian kernel. We will elaborate this more in revision. For spatially-varying blurring, while we empirically observed it converges, the theoretical analysis of the eigenvalues of a spatially-varying blurring operator is very challenging.
>
> ------------------------------------------------------------------------------
> **Regarding the advantage of proposed method over HQS-based unfolding**
>
> The vast majority of existing unfolding algorithms for image deblurring is for non-blind image deconvolution. In non-blind image deconvolution, the blurring is assumed to be uniform which can be modeled by the convolution of a single convolution kernel $k$, and it needs to solve a linear system $y=k\otimes x+\epsilon$. As the inversion of such a convolution can be efficiently implemented via FFT, existing unfolding methods (e.g. HQS, ADMM) often use FFT to run the inversion during each iteration and then call a CNN to refine the inversion results.
>
> The defocus deblurring problem is very different from non-blind image deconvolution. It belongs to the category of non-uniform blind deblurring, which cannot be modeled by a convolution and needs to estimate both the blurring operator $B$ and clean image $x$ in $y=Bx+\epsilon $. Thus, the inversion in HQS-based or ADMM-based unfolding cannot be efficiently computed via FFT. Therefore, we proposed a fixed-point iteration scheme which does not involve the inversion of the operator $B$ but the forward operator $B$ itself only.
>
> ------------------------------------------------------------------------------
> **Regarding the artifact removal block**
>
> It is related to the discussion above. The existing unrolling methods, e.g. the HQS-based ones, explicitly run the inversion process during each iteration to have an estimate of the target image. The inversion process is not a learnable network, but a classic scheme such as FFT-based least squares. Thus, there is the need to have an explicit artifact remover in these methods.  In our approach, the inversion process is implemented by the GCM and SRAM. It is not the classic non-learnable FFT-based least squares estimator but the one containing learnable network components. Thus, our network can be viewed as some form of the inversion process with implicit built-in artifact removal.
>
> ------------------------------------------------------------------------------
> **Regarding comparison with other possible variants (e.g. HQS)**
>
> As discussed above, most unrolling methods are designed for non-blind image deconvolution. Moreover, they are for handling uniform blurring where the matrix $B$ is a perfect convolution matrix that can be diagonalized in FFT. For the matrix $B$ corresponding to spatially-varying blurring, it is challenging to do the inversion effectively in HQS, which is probably one reason why we did not see any work that unrolls the HQS to recover the image with spatially-varying blur.
>
> ------------------------------------------------------------------------------
> **Regarding “Is the proposed network trained once, and tested on different datasets? “**
>
> Yes, the proposed network is trained only on the DPD dataset. Then the trained model is tested on DPD, RTF and CUHK-BD datasets. Notice that for demonstrating the robustness to noise, in Table 2 and 3 we train our model on the DPD dataset with two noise settings respectively.
>
> ------------------------------------------------------------------------------
> **Regarding “Is the trained network generalizable to defocus images captured by different cameras?”**
>
> Yes.  Recall that the cameras are different on different benchmark datasets and our model trained on one dataset performs better than other methods on another dataset in the experiments. Thus, the results in the paper have shown that our trained network also generalizes better than other methods on defocus images captured by different cameras.
>
> --------------------------------------------------------------------------------
> **Regarding analysis of the limitations**
>
> While our approach has achieved SOTA results on existing datasets, the performance on some severely blurred images remains unsatisfactory, as most information is lost by severe blurring. Building a dataset that covers more severely blurred images and developing effective methods to recover such images remain to be further studied. We will add this discussion in Conclusion.

---

> > ### Comment · Reviewer_SpFs · 2021-08-23
> > **Comments on rebuttal**
> >
> > Thanks for clarifying the concerns on convergence and comparisons with HQS-based methods. The responses to convergence issue is reasonable to me. However, the responses to the comparisons with HQS-based method (especially with artifact removal block) are not convincing to me. First, I agree that the problem of interest in this work is on the non-uniform deblurring task, which is less tackled in unrolling-based method. However, it is not hard to extend the HQS-based unrolling method to non-uniform deblur problem, because the FFT-based solver can be replaced by the iterative solver for the deconvolution sub-problem, possibly by implicit differentiation or further unfolding the iterative solver. Second, though GCM and SRAM have parameters to learn, however, to my understanding, their parameters are quite limited, mainly for the learning of blur kernel, which should be less effective in removing artifacts compared with the learnable de-artifacts network block. In this sense, I suspect that the proposed framework may not be optimally designed, and introducing the de-artifacts network block may further improve the results.
> >
> > As a summary, I still tend to retain my previous rating.

---

> > > ### Author Response · Authors · 2021-08-24
> > > **Responses to the comments at the 2nd round**
> > >
> > > Thanks for the reply. We would like to point out that, while our method cannot be mathematically proven to be an optimal NN for defocus deblurring, our empirical experiments showed it is the best performer among all existing solutions. There is certainly the room for further improvement, just like all works do. We believe that the possibility of further improvement does not impact the value of our work.
> > >
> > > Indeed, the effectiveness and efficiency are two main concerns when we proposed the fixed-point iteration for non-uniform deblurring. The HQS-based scheme has its issue on computational efficiency. When addressing the inversion of the non-uniform blurring operator by calling an iterative solver, embedding an iterative solver into each iteration of HQS will lead to much higher computational cost. In addition, training via back-propagation will see a significant increase of computational cost, as each block of the unrolled HQS contains several iterations which significantly increase the cost for gradient calculation.
> > >
> > > We ran the experiments to test the HQS-based unrolling scheme as the reviewer suggested. Following the HQS+CG NN for non-blind deblurring [a], the conjugate gradient (CG) iteration is used for the inversion (as the non-blind deblurring did) and a DnCNN-like denoiser block is used for artifact removal block. The blur operator defined by our SRAM+GCM is inserted into the HQS+CG framework. For fair comparison, we also adopt the same multi-scale scheme and also share the weights of blocks across scales. The number of CG iterations in each scale, denoted by P, is set to10,15,20,25, as such values are often used in existing CG-unrolling-based NNs for image deblurring.  See following table for the comparison on efficiency. It can be seen that such an HQS-CG unrolling scheme is much slower than our fixed-point iteration.
> > >
> > > | Method                        | #Parameters(Million)$\downarrow$  | FLOPs(Billion)$\downarrow$       | Time(Second)$\downarrow$|
> > > | :------------                 |:-------------:                    |:-------------:                   |:-------------:          |
> > > | HQS+CG ( P=10)                |3.26                               |4994.3                            | 0.308|
> > > | HQS+CG ( P=15)                |3.26                               |7264.5                            | 0.485|
> > > | HQS+CG ( P=20)                |3.26                               |9534.7                            | 0.611|
> > > | HQS+CG ( P=25)                |3.26                               |11804.9                           | 0.824|
> > > | Ours                          |1.41                               |603.5                             | 0.040|
> > >
> > > In addition, we also ran the experiments to study whether the learnable artifact removal block from [a] benefits our method. The artifact removal blocks are attached to the output of each scale of our model and share weights across scales (same as SRAM), and the loss is added to both the output of each SRAM and the output of each artifact removal block. We retrained the new model for fair comparison. See the following table for the results. The performance gain by introducing such an artifact-removal block to our NN is little, while it brings a noticeable decrease on the computational efficiency.
> > >
> > > | Method | PSNR(dB)$\uparrow$    | SSIM$\uparrow$    | FLOPs(Billion)$\downarrow$    |
> > > | :------------            | :-------------:       | :-------------:             |:-------------:                |
> > > | Ours + Artifact Removal | 25.49                 |  0.789                                    |1984.1                         |
> > > | Ours                       | 25.47                 |  0.789                                    |603.5                          |
> > >
> > > Overall, an HQS-based iteration is much less computationally efficient than the proposed method and the possible gain by adding an artifact removal block is negligible.
> > >
> > > We sincerely thank the reviewer's feedback, and wish our points can be considered for the final evaluation of our work.
> > >
> > > [a] End-to-end interpretable learning of non-blind image deblurring. ECCV 2020.

---

### Official Review · Reviewer_1KQg · 2021-07-16

**Rating:** 7
**Confidence:** 4

**Summary:**

The paper proposes a neural network architecture for deblurring images affected my single image defocus blur. The architecture is inspired by the notion that (a) the blur kernel at any pixel for defocus blur can be well approximated as a combination of Gaussian kernels with different widths, and (b) given knowledge of the blur kernel, the deblurred image is given by the fixed point of a repeated filtering operation. These notions are combined with multi-scale reasoning, and leads to a network that performs well in practice while being very light-weight and with fewer number of parameters. This is demonstrated by experiments and ablations.


**Main Review:**

Overall, the proposed architecture is novel, well-motivated, and the experiments are fairly convincing. I'm leaning towards acceptance, but hope the authors will respond to/address the following issues:

1. Calling the kernel model a Gaussian Scale Mixture (GSM) is very confusing. This is because Gaussian Scale Mixture *distributions*, as probabilistic models of image gradients/wavelet coefficients, were a very popular, successful, and widely used approach to traditional image restoration. In this context, the "GSM" refers to the approximation of isotropic kernels as sums of Gaussian of different widths (scales). I strongly recommend the authors consider calling this "Gaussian Kernel Mixture" or something else.

On a similar note, it may be worth calling \gamma's mixing weights or some such instead of attention masks (because again, attention has different connotations in neural network architectures).

2. The description of the fixed point iteration as motivation is great, but while in Sec 1.2, this is applied to multiple iterations at the same scale, the actual architecture also simutaneously keeps increasing the scale at which the method is applied. This makes things a little bit confusing. Perhaps one could write the architecture as K scales and R iterations per scale, with total number of "iterations" = K x R, and that in this case, the authors chose to go with R=1. And that brings up another interesting ablation: is multi-scale really helpful, or would the network have achieved equivalent accuracy with fewer (or even a single) scale, and multiple blocks of GCM/SRAM per scale.

3. Some of the specific details of the architecutre are hard to decipher from the figures. For one, I think there's a typo in equation (8)---I imagine it should be y_t + not y +. Also, it isn't clear what if any weights are shared across different scales. The GCM section talks about a fixed set of kernels, but it isn't clear whether the \sigma values are at the downsampled or original resolution. And in general, for both the GCM and SRAM modules, it would be good to have text describing the precise input/output relationships in the supplementary (it's fine to use existing architecture blocks such as Conv-LSTM as "functions", but deciphering the architecture is a bit hard from just the figures).

4. What's the difference between #parameters and model-size in Table 1?

5. It isn't immediately clear how the attention maps are closely related to the defocus map in Figure 3.


**Time Spent Reviewing:**

2

---

> ### Author Response · Authors · 2021-08-10
> **Responses to the comments from Reviewer 1KQg**
>
> We sincerely thank the reviewer for his/her constructive comments.
>
> ------------------------------------------------------------------------------
> **Regarding the suggested terms “Gaussian Kernel Mixture” and “$\gamma$'s mixing weights”**
>
> Thanks a lot for the suggestions. We will adopt the suggested terms “Gaussian Kernel Mixture” and “$\gamma$'s mixing weights” to avoid possible confusion.
>
> -----------------------------------------------------------------------------
> **Regarding the experiments on the number of scales and the number of iterations**
>
> As computational efficiency is also one key goal to achieve. we tried the suggested setting with K=1 and R=3, which leads to the same model size as the original one. Please see below for the comparison.
>
> | Method | PSNR(dB)$\uparrow$ | SSIM$\uparrow$ | LPIPS$\downarrow$ | #Parameters (Million)$\downarrow$ | Time(s) $\downarrow$ |
> | :------------: | :-------------:  | :-------------: | :------------: | :-------------: | :-------------: |
> | K=1, R=3   | 25.11           |  0.773         | 0.237      | 1.41   | 0.069          |
> | Original     | **25.47**           | **0.789**          | **0.219**  | 1.41      |  **0.040**        |
>
> The study showed that using single scale with multiple iterations noticeably decreases the performance and increases the inference time. Thus, our coarse-to-fine estimation scheme is a better implementation for high performance and low computational cost.
>
> -----------------------------------------------------------------------------
> **Regarding specific details**
>
> (i) Yes, the typo in Equation (8) will be corrected.
> (ii) In GCM, the kernels and their parameters are fixed at the original resolution across all scales.
> (iii) Thanks for the suggestion. We will add the text description as suggested.
>
> -----------------------------------------------------------------------------
> **Regarding the difference between #parameters and model size**
>
> The former is about how many parameters in the model, while the latter is file size of the model saved on hard disk. They are two often-used metrices for reflecting how large a model is.
>
> -----------------------------------------------------------------------------
> **Regarding Figure 3**
>
> Figure 3 is for illustrating the consistency between the main structure in the attention maps and the defocus map. We will add more intuitive examples for examination in the revision.

---

> > ### Comment · Reviewer_1KQg · 2021-08-28
> > **Thanks!**
> >
> > Thanks for the clarifications. Couple of points:
> >
> > 1. I can believe that multiple scales with single iteration is better than multiple iterations at each scale. It might be good, for completeness, to show multiple iterations per scale with multiple scales as well (even if it doesn't do much better than single iteration-multiple scales and takes more FLOPs, it's a good experiment to add for completeness).
> >
> > 2. I don't think Figure 3 necessarily needs more examples. Rather, it would be good to add text to the caption specifically pointing to the similarities, rather than leave it to the reader to interpret.
> >
> > Otherwise, I retain a favorable opinion of the paper.

---

> > > ### Author Response · Authors · 2021-09-04
> > > **Responses at the 2nd round**
> > >
> > > We sincerely thank the reviewer for the comments this round.
> > >
> > > Please see followings for the results using multiple iterations per scale with multiple scales. It brings slight improvement on certain metrics. We will add such results in revision.
> > >
> > > | Method            | PSNR(dB)$\uparrow$    | SSIM$\uparrow$    | LPIPS$\downarrow$ | #Parameters (Million)$\downarrow$ | Time(s) $\downarrow$  |
> > > | :------------   | :-------------:       | :-------------:   | :------------:    | :-------------:                   | :-------------:       |
> > > | K=1, R=3          | 25.11                 |  0.773            | 0.237             | 1.41                              | 0.069                 |
> > > | K=2, R=3          | 25.13                 |  **0.789**        | 0.205             | 4.22                              | 0.122                 |
> > > | K=3, R=2          | **25.53**             |  0.773            | **0.201**         | 2.81                              | 0.081                 |
> > > | K=3,R=1 (Original)          | 25.47                 |   **0.789**              | 0.219             | **1.41**                          |  **0.040**            |
> > >
> > > Regarding Figure 3, we will revise the caption as suggested and add some annotations on the figure for better illustration.

---

### Official Review · Reviewer_SBT5 · 2021-07-16

**Rating:** 6
**Confidence:** 4

**Summary:**

The authors propose an attentive GSM network for defocus deblurring, motivated from the unrolling of a fixed-point iteration.
Experiments show that the proposed method in not only lightweight, but also outperforms existing defocus deblurring methods.



**Limitations And Societal Impact:**

yes

**Main Review:**

The paper is well-written and organized.
Based on a fixed-point iteration, the authors propose a new formulation of deblurring process.
The authors combine coarse-to-fine progressive estimation and unrolled deblurring process into a scale-recurrent attention mechanism.
The experiments show the superior of the proposed method in terms of quality, model size, Flops, and running time.
It would be better if the authors could provide demo testing code for the method.

After rebuttal

Thanks for providing the feedback. I also read comments from other reviewers and like to keep my original rating.


**Time Spent Reviewing:**

11

---

> ### Author Response · Authors · 2021-08-08
> **Responses to the comments from Reviewer SBT5**
>
> We sincerely thank the reviewer for his/her constructive comments.
>
> **We promise that all the training and testing codes for reproducing all results will be definitely released to the public on GitHub upon the acceptance of the paper.**

---

> > ### Comment · Reviewer_SBT5 · 2021-08-23
> > **After rebuttal**
> >
> > Thanks for providing the feedback. I also read comments from other reviewers and like to keep my original rating.

---

### Official Review · Reviewer_mowT · 2021-07-19

**Rating:** 3
**Confidence:** 5

**Summary:**

This paper focuses on addressing the challenge of single-image defocus deblurring.
This work is based on the GSM (Gaussian scale mixture) theory to construct a deblurring pipeline with the DNN (deep neural network).
This paper uses extensive experiments to verify the effectiveness and robustness of the model.


**Main Review:**

[Strengths]

1)	Gaussian mixture methods are usually used to fit unfriendly data distributions, and this is the first case I have visited where the use of both the Gaussian kernels and the coefficient  matrices to accomplish fitting arbitrary data distributions, may be a very bold paradigm of innovation.

2)	A large number of experimental comparisons have provided the evidences to support the validity of this work.


[Weaknesses]

Although the proposed model is observed to be valid in terms of quantitative and qualitative results, I still have doubts about the contributions of this work.

1) In L144-145, this work assigns equal weights to the loss at each scale of AGSMNet. In this regard, as far as I know we are more concerned about the clarity of the output of the last scale, and it is natural to think that the last scale los should be assigned the largest weight.

2) In L160-161, this work uses the APU to generate coefficient maps, APU is based Conv-LSTM. This method still seems to have a high computational, so why not consider using the encoder of the transformer to replace it? I'm looking forward to seeing the new ablation experiments.


3) In Table1, I am still troubled by why the proposed model has the least FLOPs, but  process of the GPU is not fast.

4) The Gaussian kernel itself has been applied to the denoising task, so the robustness of the model is necessarily excellent and the robustness consideration of the experimental part seems unnecessary.

5) I think the biggest problem of this paper is still the limited innovation, which seems to box up multi-scale, long range dependence, dual path and other methods to build the model.


**Time Spent Reviewing:**

4

---

> ### Author Response · Authors · 2021-08-10
> **Responses to the comments from Reviewer mowT**
>
> We sincerely thank the reviewer for his/her constructive comments.
>
> ------------------------------------------------------------------------------
> **Regarding the innovation**
>
> We **respectfully disagree with** the assertion that our work simply puts many existing modules together without innovation.
>
> While the element blocks of the proposed network are not new, our innovation lies in architecture design, which is motivated from a new model of defocus blurring and a fixed-point iteration, **as Reviewer 1KQg commented.** It includes
>
> (i) A new model using Gaussian Kernel Mixture for pixel-wise isotropic defocus blur kernel, which is more accurate than the existing ones.  We would clarify that the usage of Gaussian Scale Mixture is **not** for modeling unfriendly distributions as suggested in [Strengths], but for approximating the defocus kernel. The confusion can be caused by the usage of the term "Gaussian Scale Mixture". We will switch to “Gaussian Kernel Mixture” as suggested by Reviewer 1KQg.
>
> (ii) A network structure derived from the unrolling of a fixed-point iteration with the proposed defocus kernel model, which achieves better computational efficiency. The GCM and SRAM, as well as their inclusion, have strong inspirations from the proposed defocus blur model. In other words, we are not just stacking all the network modules together to improve performance but designing a new network architecture which effectively exploits the characteristics of defocus blurring.
>
> ------------------------------------------------------------------------------
> **Regarding "The Gaussian kernel itself has been applied to the denoising task, so the robustness of the model is necessarily excellent and the robustness consideration of the experimental part seems unnecessary."**
>
> When being applied to the denoising task, the Gaussian kernel is used as a low-pass filter to convolve the image to suppress noise. In our problem setting, Gaussian kernels are **not used as the denoising filters**, but used as the basic elements to model the defocus blur kernel for deblurring. As the deblurring process may be sensitive to noise without special treatment, the experiments on noise robustness can clear off possible concerns on such an issue.
>
> ------------------------------------------------------------------------------
> **Regarding why the proposed model has the least FLOPs but its processing on GPU is not the fastest**
>
> When running on CPU with single thread, fewer FLOPs generally lead to shorter inference time. Thus, our method achieved both the least CPU running time and the least FLOPs. However, when running on GPU, there are many factors in implementation that can influence the  inference time, such as parallel programming tricks, sync overhead, memory access cost, number of element-wise operations, and environment support; see [a] for more details.
>
> Our implementation has not been fully optimized for the GPU computation yet. Even so, its GPU running time is still much less than DPDNet. We will continuously refine our GPU implementation after publishing our work on Github.
>
> [a] Practical Guidelines for Efficient CNN Architecture Design, ECCV 2018.
>
> ------------------------------------------------------------------------------
> **Regarding replacing Conv-LSTM with transformer encoder**
>
> Thanks for the suggestion. The suggested study is conducted by replacing the Conv-LSTM with the transformer encoder [b] and configurating it to have a similar number of parameters as the implemented Conv-LSTM. In the experiments, we **did not observe any improvement on inference time or performance**. See below for the results.
>
> | Model         | PSNR(dB)$\uparrow$    | SSIM$\uparrow$        | LPIPS$\downarrow$     | #Parameters (Million)$\downarrow$| Time(s) $\downarrow$  |
> | :------------:| :-------------:       | :-------------:       | :------------:        |:-------------:     | :-------------:       |
> | Encoder of Transformer | 24.91        |  0.762                | 0.236                 | 1.98                   | 0.058                 |
> | Conv-LSTM      | **25.47**            | **0.789**             | **0.219**             | **1.41**                |  **0.040**            |
>
> There are two reasons why there is no gain in inference time. (i) The implemented Conv-LSTM has only three units and its channel number is small. Thus, the advantage of transformer encoder on computational time is not significant and it even becomes worse. (ii) The significant part of the computational cost also comes from other network parts, including the sequential processing in GCM over scale, and the element-wise products in Equation (8), as sequential operations and element-wise operations can be slow on current GPU support [a].
>
> [a] Practical Guidelines for Efficient CNN Architecture Design, ECCV 2018.
>
> [b] Pre-trained image processing transformer, CVPR 2021.
>
> ------------------------------------------------------------------------------
> **Regarding assigning non-equal weights in the loss**
>
> Thanks for the comment. Yes, using increasing loss weights from the roughest scale to the finest scale may give some performance improvement. However, our empirical studies showed that the improvement is very minor, e.g., the PSNR improvement introduced by setting the weights to (0.75, 1, 1.25) is less than 0.014dB over all test data. Thus, we simply use the same weight (set to 1) across scales. This tuning-free setting does not lower the value of our work. With such a setting, our network still achieved SOTA results. This not only demonstrated the effectiveness of our network architecture for defocus deblurring, but also showed the insensitivity of these weights to the performance.

---

### Decision · Program_Chairs · 2021-09-28

**Decision:**

Accept (Poster)

**Comment:**

The paper received mixed reviews (3,5,6,7). One reviewer is particularly enthusiastic about the paper.  One reviewer was negative but did not participate in the discussion. His/her comments have been well addressed in the rebuttal and his/her final score of 3 has been ignored from the decision process. Two reviewers have borderline ratings (5,6), and the reviewer with score 5 agreed in the private discussion that it would be fine if the paper is accepted.

Overall, the AC believes that the paper introduces interesting ideas and recommends acceptance.  Nevertheless, Reviewer SpFs raises relevant questions about why the method works and why it makes sense. These should be taken into account in the final version of the paper. You will find below part of the private discussion, which will clarify these concerns and give some hints about how they can be addressed.

***************************
**Reviewer SpFs:**
In the second round of responses, the authors conduct experiments to compare with HQS-unrolling based method, and the conclusions are that the HQS is with higher computation cost and the additional artifacts removal block did not improve much. I appreciate these additional comparisons and it is still a mystery to me on the reason why this propose approach works well even without using artifacts removal block, because the artifacts are commonly serious in deblur / non-uniform deblur tasks. If not explicitly penalize the artifacts, purely relying on deconvolution (implemented by fixed point iterations with learned parameters for blur kernel modeling) can hardly work well. Moreover, the paper did not provide accuracy results of HQS-based methods. However, maybe I miss something / mis-understand on something. I stand on the borderline and is OK with both rejection / acceptance.

**Reviewer 1KQg:**
So, my reading of the paper is a bit different. I think this paper's a little different from HQS methods where the iterations are explicitly minimizing a deblurring objective (i.e., estimate * kernel = observation) with a denoising network as the "prior".

The proposed method is just using fixed point iterations as the 'motivation' for it's architecture design: what they're proposing is essentially just a new kind of network architecture that's trained end-to-end (with the benefit that it's very light-weight and efficient). In the literature, we've seen that direct CNNs trained end-to-end with a reconstruction (and other) losses can achieve state-of-the-art performance.

This paper's method is sort of in that "family" of methods, and so perhaps it's not surprising it can do well without an explicit artifact removal step. In some sense, the fact that the underlying fixed point iterations didn't have an artifact penalty doesn't matter because the kernels are all being learned and with the objective of maximizing the final output's quality. What's novel about the paper is that compared to prior end-to-end networks, this one's architecture is efficient and has this nice motivation from classical deblurring methods.

**Reviewer SpFs**: Thanks, I agree that the proposed network is lightweight and efficient. Regarding the comment "fixed point iterations didn't have an artifact penalty doesn't matter because the kernels are all being learned and with the objective of maximizing the final output's quality", my understanding is as follows. Even the blur kernel is accurately estimated, the deconvolution step will still introduce artifacts (please consider the non-blind image deblurring task). Since this proposed network may not have enough capacity to remove artifacts, to minimize the reconstruction loss, the network may be trained to be conservative and insufficiently remove blurs (because sufficiently removing blur may introduce artifacts and increases the reconstruction error). I maybe wrong, but these are my personal concern.

**AC**: I agree that we do not understand perfectly why the approach works and that there is probably a regularization effect due to the fact that the fixed point iterations do not necessarily converge to a fixed point (in the same way as early stopping can induce regularization in optimization methods). The exact nature of the regularization is unknown, but it does not necessarily prevent it from working since all parameters are learned end to end. I will encourage the authors to think about it and include a discussion in the paper.


**Consistency Experiment:**

NeurIPS has a long history of experimentation. In 2014, NeurIPS ran an experiment in which 10% of submissions were reviewed by two independent committees to quantify the randomness in the review process. This year, we repeated a variant of this experiment to see how the quality of the review process has changed over time.  This paper was part of the experiment and was therefore assigned to two committees (consisting of reviewers, an Area Chair, and a Senior Area Chair) that reached independent decisions.  If both committees made the same recommendation, this recommendation was followed. If a single committee recommended acceptance, the paper was accepted (with the exception of a few cases in which the other committee identified what we considered a fatal flaw, e.g., an error in a key result).

This copy’s committee reached the following decision: **Accept (Poster)**

The other committee assigned to the paper recommended **Reject**.  You can find the other set of reviews, along with any follow up discussion with the authors here:
https://openreview.net/forum?id=kSR-_SVzDR-